# Influence of Green Finance on Ecological Environment Quality in Yangtze River Delta

**DOI:** 10.3390/ijerph191710692

**Published:** 2022-08-27

**Authors:** Meili Tang, Jia’ni Ding, Haojia Kong, Brandon J. Bethel, Decai Tang

**Affiliations:** 1School of Marxism, Nanjing University of Information Science & Technology, Nanjing 210044, China; 2School of Management Science and Engineering, Nanjing University of Information Science & Technology, Nanjing 210044, China; 3School of Economics and Management, Nanjing University of Science & Technology, Nanjing 210094, China; 4School of Marine Sciences, Nanjing University of Information Science & Technology, Nanjing 210044, China

**Keywords:** green finance, ecological environment quality, Yangtze River Delta

## Abstract

Along with the development of society and the deepening contradiction between economic growth and natural resources, green finance has attracted more and more attention. As an area of great strategic significance in China’s modernization, the development of green finance can improve the quality of its ecological environment and find new economic growth points for it. Based on the index system of pressure state response and while considering the scientific nature and desirability of the indicators, this paper selects 12 indicators to construct an index system of eco-environmental quality. It uses the entropy method to calculate the level of eco-environmental quality. Then, three control variables are selected, and the difference-in-difference model is used for empirical analysis. It is found that green finance positively affects the ecological environment quality of the Yangtze River Delta. In addition, the level of opening to the outside world and the level of economic development also have a positive effect on the quality of the ecological environment to a certain extent. Still, the impact of industrial structure on it is negative. Therefore, this paper puts forward some suggestions for strengthening the disclosure of green financial information, paying attention to the concept of green development and strengthening regional cooperation and exchange to promote the development of green finance further and promote the coordination of economic development and ecological protection in the Yangtze River Delta.

## 1. Introduction

Since the completion of reform and opening up, China’s economic development has accelerated, but at the same time, ecological and environmental problems have become increasingly serious [1]. A series of problems such as air pollution, water pollution, and energy shortage is emerging, and the pressure on the ecological environment is increasing rapidly. Along with the popularization of the concept of sustainable economic development, China is paying more and more attention to ecological environmental quality [2]. The traditional mode of economic development at the cost of environmental damage is in urgent need of change, and the pressure of economic structural transformation is surging.

The development of green finance in China started late, but in recent years, the research results and policy support for green finance have increased. In August 2016, seven ministries and commissions issued an important document on green finance, aiming to optimize the allocation of market resources, promote ecological civilization, support sustainable economic development, and provide guidance for establishing China’s green finance system. At present, promoting rational adjustment of industrial structure and green economic transformation is China’s important direction and goal. The fifth Plenary Session of the 19th Central Committee of Communist Party of China (CPC) even proposed that green finance plays a role in supporting the harmonious coexistence between man and nature and emphasized the important role of green finance in green development. In July 2021, the Environment Chamber of Commerce of the All-China Federation of Industry and Commerce held a forum in Chengdu. It proposed that China’s ecological civilization construction is in a critical period and carbon reduction will be the strategic direction for the future, which has attracted widespread attention from the society.

The Yangtze River Delta (YRD) mainly consists of Shanghai, Jiangsu, Zhejiang, and Anhui (Figure 1). In the world’s advanced modern service industry and manufacturing industry, it is a world-class metropolis group and an important manufacturing center [3]. Given the change of policy orientation and the increase of economic development pressure, urban agglomeration in YRD needs to find new economic growth points, improve energy utilization efficiency and environmental pollution control ability, and develop a green economy. In 2021, the YRD Eco-green Integrated Development Demonstration Zone issued a notice, making the YRD a pilot zone for green financial products and green financial innovation. As a financial activity to coordinate economic development and environmental protection, green finance is playing an increasingly indispensable role in realizing green economic development in the YRD, and the government should pay more attention to it. At present, the YRD still has a problem of imbalance between economic development and ecological environment protection. Green finance is the key to realizing the adjustment of industrial structure and green economic transformation, and also an important guarantee for promoting the harmonious coexistence between man and nature and the improvement of ecological environment quality. Therefore, the development of green finance is of great significance.

Unlike traditional finance, green finance pays more attention to the coordinated development of the economy and environment and promotes the construction of ecological civilization. Scholtens [4] studied the transmission mechanism between finance and sustainable development and solved resource and environmental problems through the optimal combination of financial instruments. Marcel [5] emphasizes that the concept of green finance must be understood from the perspective of the green development of the financial sector. Anderson [6] proposed that green finance is a new mechanism for banks to finance and issue loans for the environmental industry. Wang and Zhi [7] proposed that green finance is a financial model that combines environmental protection with economic benefits.

This paper attempts to study the impact of green finance on the ecological environment quality in the YRD. Based on the index system of pressure state response (PSR) and while considering the scientific nature and desirability of the indicators, we selected 12 indicators to construct an index system of eco-environmental quality and used the entropy method to calculate the level of eco-environmental quality. Then, three control variables were selected, and the difference-in-difference (DID) model was used for empirical analysis. The innovation of this study has the following two points: first, although the relationship between the green financial and ecological environment quality has been studied widely, the related research for a particular economic area is less extensive. This paper discusses the YRD green financial impact on the quality of the ecological environment and proposes certain innovations in terms of the research object. Second, it refers to the results of other scholars and constructs a new ecological environmental quality index evaluation system via the entropy method for Yangtze River ecological environmental quality of different YRD in 2010–2020 levels. Through the analysis of the YRD green finance and panel regression model, the relationship between the ecological environment quality of found green finance has a significant positive effect on the ecological environment quality. Compared with the existing literature, some innovations in index construction and research methods are presented.

## 2. Literature Review

As for the importance of green finance development and enterprise development, Scholtens and Dam [8] found that financial institutions carrying out the “Equator Principle” had a higher social responsibility. By comparing green bonds with non-green bonds, Gianfrate and Peri [9] found that green bonds are more convenient in finance and have advantages for corporate issuers, which will play an important role in a green economy. Xu et al. [10] found that green finance has a significant positive effect on the green performance of enterprises. Zhang et al. [11] found that developing green finance is more conducive to seizing opportunities and implementing product innovation. In terms of social and economic development, Shi and Geng [12] believe that green finance positively impacts both environmental protection and economic development. Sachs et al. [13] believe green finance is significant for sustainable development and energy security. Zhang and Wang [14] constructed an evaluation system for green finance development and found that green finance can promote sustainable energy development. Wang et al. [15] used the gaussian mixture model (GMM) and grey correlation method to prove that green finance can promote industrial structure optimization and sustainable development of a green economy. In terms of international development, Loukil and Jarboui [16] analyzed the application of the “Equator Principle” in 78 international banks. They found that investment in the green industry would create more market value and promote sustainable economic development. MacAskill et al. [17] believe that green bonds are becoming an influential financing mechanism for climate change mitigation.

Based on the theory of ecology, ecological environmental quality is a compound ecosystem reflecting natural resources and human existence. For the ecological environmental quality evaluation system, David [18] first established the random sample partition (RSP) model to evaluate ecological environmental quality, namely the PSR index system, which became an important reference for later scholars to construct the index system. Han et al. [19] assessed the ecological risk of 13 sample cities in the Beijing–Tianjin–Hebei region based on the RSP conceptual model. Derakhshannia et al. [20] evaluated the quality of Iran’s aquatic ecosystem based on this model. Das et al. [21] used the RSP model to discuss the ecological environmental quality of wetlands in West Bengal, India. After the progress and gradual popularization of remote-sensing technology, scholars increasingly apply the environmental index based on remote sensing. Shan et al. [22] found that the Remote Sensing Ecological Index (RSEI) was better than Ecological Index (EI) for reflecting changes in ecological environment quality. Wu et al. [23] used remote-sensing image data, combined with analytic hierarchy process (AHP), variation coefficient method, and geographic information science (GIS) to evaluate the ecological environment quality in western Chongqing.

Crossman and Krueger [24] proposed an inverted U relationship between economic growth and environmental pollution in studying the relationship between green finance and environmental quality. Liddle and Messinis [25] verified the inverted U relationship between economic development and environmental quality. Liu et al. [26] measured the development level of green finance in all provinces of China and confirmed the role of green finance in improving environmental quality. Zhou et al. [27] selected industrial soot, carbon dioxide, solid waste, and other indicators as environmental variables to build a green development index and found that green finance can improve environmental levels. Batrancea [28] found that the financial sector financing green investment can effectively promote economic growth and reduce global warming and climate change. Huang and Zhang found [29] that the green finance pilot zone positively reduces environmental pollution. The more serious the environmental pollution, the better the effect of green finance. Tao and Zhang et al. [30,31] studied the coupling coordination degree of green finance and ecological environment in China and found that it was at a medium and low level with an upward trend, presenting a spatial layout of high in the east and low in the west.

Through the above literature review on the meaning of green finance, the importance of developing green finance, the evaluation system of ecological environment quality, green finance, and ecological environment quality, it was found that domestic and foreign scholars have conducted relatively mature research in related fields. From the perspective of the importance of developing green finance, domestic scholars pay more attention to developing financial institutions and sustainable economic development. In recent years, scholars have paid more and more attention to the role of green finance in the ecological environment and carried out research at all levels of the country, province, and county. The main research methods include the comprehensive index method, analytic hierarchy process, etc., which are cross-used with remote-sensing technology. However, there are still some deficiencies in the existing research. First, in terms of the impact of green finance on ecological environment quality, most studies are currently conducted at the provincial and municipal levels, and there is a lack of research on the level of a certain economic region. Second, because different scholars choose different research samples and evaluation indicators, there are errors in the results obtained.

This paper takes green finance and ecological environment quality in the YRD region as the main research objects. It refers to the existing research of scholars, tries to build an indicator system of ecological environment quality, and uses a panel regression model to analyze its relationship with green finance. This study is helpful in analyzing the impact of green finance on the ecological and environmental quality of the YRD, improving the relevant theories of green finance, and promoting the coordinated development of the economy and environment.

## 3. Model Construction and Data Explanation

### 3.1. Theoretical Mechanism

Along with the development of society, China’s awareness of environmental protection is becoming stronger and stronger, and the government has issued a series of policies conducive to ecological environmental protection. China’s air quality has continued to improve, soil environmental risks have been basically controlled, and the aggravation of soil pollution has been initially curbed, according to a communique on the state of the ecological environment. Therefore, based on the above situation, Hypothesis 1 is proposed.

**Hypothesis** **1** **(H1).**
*The eco-environmental quality level within the YRD is generally on the rise.*


As a hot topic in recent years, green finance can support environmental benefits on the one hand and promote sustainable economic activities on the other hand. The government has issued policies supporting green finance in the YRD region, where the Yangtze River Economic Belt meets the Belt and Road Initiative. Based on the above situation, Hypothesis 2 is proposed.

**Hypothesis** **2** **(H2).**
*Green finance promotes the eco-environmental quality of the YRD.*


### 3.2. Modeling

#### 3.2.1. Entropy Method

In order to test Hypothesis 1, an index system of eco-environmental quality based on PSR index system was constructed, and the comprehensive index of ecological environment quality was calculated by the entropy method.

The specific steps are as follows:

Step 1: Establish the following eco-environmental quality index system based on PSR index system, as shown in Table 1. Pressure indicators include industrial wastewater emissions, industrial soot emissions, sulfur dioxide emissions, and general solid waste production. State indicators include per capita water resources, per capita urban park area, per capita road occupancy, forest coverage rate, and green coverage rate of built-up areas. The response indicators include urban sewage treatment rate, the comprehensive utilization rate of industrial solid waste, and the harmless treatment rate of domestic waste.

The ecological environment involves resources, environment, economy, society, and other fields, so the design of the index system must follow certain principles. PSR index system is a framework system developed by the organization for economic cooperation and development and the United Nations Environment Programme to study environmental issues. The PSR index system theory is used to reflect the interaction between humans and the environment and is respected and widely used by many scholars at home and abroad. Based on the characteristics of water and soil resources, environmental climate, human activities, and economic development in the Yangtze River Delta and the existing resource and environmental constraints, this paper follows the principles of the availability, typical representativeness, and scientific rationality of each index data point. It focuses on selecting 12 indicators based on the PSR index system to build the YRD eco-environmental quality evaluation index system, fully consider the complex relationship of the interaction between the regional ecological environment quality evaluation factors, and compare with the existing relevant research results. In the index system, the pressure on the ecological environment is reflected in the emission of pollutants, the state of the ecological environment is reflected in the state of regional ecological resources, and the response measures taken by human beings to deal with the changes of the ecological environment are reflected in the dimension of pollution control.

Step 2: The entropy method is used to calculate the ecological environment quality level of the YRD. The specific calculation process is as follows:

First, normalize the value of each indicator and clarify the positive and negative properties. If the impact of an indicator on the overall evaluation is positive, Formula (1) is used. Otherwise, Formula (2) is used for processing. Obtain the processed value $X_{ij}$ and $X_{ij}^{\prime }$. In the calculation process, *i* represents a certain year, *j* represents a certain index, X represents the current value, and X*_max_* and X*_min_* indicates the maximum and minimum value of an indicator.
(1)$$X_{ij}=\frac{X_{max}-x}{X_{max}-X_{min}}$$
(2)$$X_{ij}^{\prime }=\frac{x-X_{min}}{X_{max}-X_{min}}$$

Second, calculate the proportion of the *jth* index in the whole population in the *ith* year, represented by $\lambda_{ij}$.
(3)$$\lambda_{ij}=\frac{X_{ij}}{\sum_{i=1}^{m}X_{ij}}$$

Step 3: Calculate information entropy $E_{j}$ and use m to represent the total number of sample YRD.
(4)$$E_{j}=-\frac{1}{lnm}\sum_{i=1}^{m}\left(\lambda_{ij}\times ln\lambda_{ij}\right),\;0\;\leq \;E_{j\;}\leq \;1$$

Step 4: Calculate information entropy redundancy $d_{j}$.
(5)$$d_{j}=1-E_{j}$$

Step 5: Calculate the weight of each indicator $W_{j}$.
(6)$$W_{j}=\frac{d_{j}}{\sum_{i=1}^{m}d_{j}}$$

Step 6: Multiply the weight $W_{j}$ (6) of each indicator by the normalized value $X_{ij}$ (1) or $X_{ij}^{\prime }$ (2) to obtain the comprehensive environmental quality index. 

#### 3.2.2. Difference-in-Difference Model

This paper uses a difference-in-difference (DID) model to study the impact of green finance on the ecological environment quality of the YRD. The DID model is one of the effective methods to test the effect of policy implementation. Its core idea is to peel off the net effect of policy implementation by establishing the model and effectively controlling the differences between research objects before and after the event. The specific approach is to find an experimental group and a control group. At a certain time, the experimental group is impacted by a policy, while the control group is not impacted; the experimental and control group’s economic development changes before and after this time are compared. If the change in the experimental group is significantly greater than that in the control group, the policy impact is significant. According to the basic idea of the DID model, this paper selects a certain time as a node. It sets an area where green finance is implemented as the experimental group and an area where green finance is not implemented as the control group. The built DID model is shown below.
(7)$${\mathrm{EQ}}_{it}=\beta_{0}+\beta_{1}treat_{i}\times time_{t}+\beta_{2}{\mathrm{GF}}_{it}+\beta_{3}\mathrm{OPEN}+\beta_{4}PGDP_{it}+\beta_{5}{\mathrm{STRU}}_{it}+\mu_{it}$$

In Equation (7), $treat_{i}$ is a grouped virtual variable. If the region implements the green finance system, the value is 1; otherwise, it is 0; $time_{t}$ is the staging dummy variable, and the value of each year after 2016 is 1; otherwise, it is 0; $\beta_{1}$ represents the effect of the experimental group after the implementation of the policy, that is, the policy effect. Where *i* represents the specific province, t represents the specific year, $\beta_{0}$ represents the intercept item, $\beta_{i}$ (*i* = 1, 2..., 4) represents the regression coefficient of each variable, *μ**_it_* represents the random error term. The description of each variable is shown in Table 2. The explained variable is EQ, representing the level of ecological environment quality, expressed by the comprehensive index of ecological environment quality calculated by the entropy method. The explanatory variable is GF, representing green finance. Each province’s investment amount in industrial pollution control is expressed in detail. In order to improve the stability of data, the logarithm of green finance is taken. The control variables are OPEN, PGDP, and STRU, where OPEN represents the level of foreign opening represented by the proportion of total imports and exports in GDP. PGDP represents the level of economic development expressed by GDP per capita. STRU represents the industrial structure, expressed by the proportion of the secondary industry’s added value in GDP.

### 3.3. Data Description

This paper selects data from Shanghai, Jiangsu, Zhejiang, and Anhui within the YRD from 2010–2020, which can be acquired from the *China Statistical Yearbook*, *China Statistical Yearbook on Environment*, *China City Statistical Yearbook,* and the *Ecological Environmental Status Bulletin*. Table 3 shows descriptive statistical data of each indicator in the ECO-environmental quality indicator system. Table 4 shows descriptive statistics for variables in the DID model.

## 4. Results and Discussion

### 4.1. Measurement of Ecological Environment Quality Level in YRD

Table 5 shows the eco-environmental quality level of the YRD from 2010 to 2020 calculated by the entropy method. From the vertical perspective, it can be found that the eco-environmental quality level of the YRD presents an overall upward trend. The comprehensive score of eco-environmental quality increased from 0.4433 in 2010 to 0.6176 in 2020, an increase of 0.1743, or 39.3%. Among them, the comprehensive score of ecological environment quality in Jiangsu has the biggest change, rising from 0.3399 in 2010 to 0.5582 in 2020, with an increase of 64.2%, indicating a significant improvement in the level of ecological environment quality. The comprehensive score of ecological environment quality in Zhejiang increased by the least, about 19.8%, which showed little change compared with the other three locations.

From the horizontal perspective, the comprehensive index of ecological environment quality in Zhejiang was the highest, reaching 0.754, ranking first among the sample YRD. The second was Anhui and Jiangsu, and the comprehensive index of eco-environmental quality was 0.5584 and 0.4425, respectively. Shanghai had the lowest eco-environmental quality index (0.3302), which may be related to Shanghai’s developed economy and high resource consumption. It follows that in recent years, the level of ecological environment quality in YRD has increased. According to the comprehensive index of ecological environment quality of all YRD, the ecological environment quality of the YRD has improved from 2010 to 2020, with the average level rising from 0.4433 to 0.6176. Presently, communities of the YRD pay more and more attention to ecological environment quality.

### 4.2. The Influence of Green Finance on the Ecological Environment Quality of YRD

Using the DID model to test the promotion effect of green finance policies on the quality of the ecological environment in the YRD region. In Table 6, (1) is the estimated result without adding control variables, (2) is the estimated result with adding control variables, $treat_{i}\times time_{t}$ and estimation coefficient of $\beta_{1}$. For the net effect of green financial policy, it can be seen that after adding appropriate control variables, green finance has a significant role in promoting the quality of the ecological environment in the YRD. In (2) treat in the column $treat_{i}\times time_{t}$, the estimation coefficient of $\beta_{1}$ is 0.034, indicating that the promotion effect of green finance policy on the improvement of ecological environment quality in the YRD is 3.4%.

Green finance positively impacts the ecological and environmental quality of the YRD. This study built a panel regression model to analyze the green financial impact on the quality of the ecological environment. The empirical results show that the green financial ecological environment quality has a significant positive influence on the development of green finance, can optimize the allocation of financial resources, guide the resources into more green environmental protection enterprise, and reduce emissions of pollutants so that green finance can promote the improvement of ecological environment quality. The level of opening to the outside world plays a positive role in promoting the quality of the ecological environment. The YRD region actively expands economic exchanges with foreign countries. It loosens various policy restrictions, which can effectively promote modernization construction and strengthen the government’s control of pollutant discharge. The level of economic development plays a positive role in promoting the quality of the ecological environment. When economic development and the development mode of production are improved, more attention is paid to the high-quality development mode, thus promoting ecological environment protection. In addition, industrial structure harms ecological environment quality. Industrial agglomeration and development increase the pressure of pollutant discharge, which increases the cost of pollution control and damages the ecological environment. In conclusion, the level of green finance, economic development, and opening to the outside world have a significant positive effect on the level of ecological and environmental quality. Therefore, the YRD should strengthen the support of green finance, practice the concept of green development, and coordinate the contradiction between economic growth and ecological environment protection.

### 4.3. Robustness Test

This paper used the placebo test to test the effectiveness of the model setting and the robustness of the estimation results. To ensure the robustness of the empirical results, we randomly selected the years before 2016, assuming 2014 as the year of green finance, and re-estimated the above benchmark model. The results are shown in Table 7. It can be seen that the estimated coefficient of the interaction term is not significant. Therefore, the impact of other potential unobservable factors on the growth of foreign trade of countries can be excluded. This counterfactual test confirms the reliability of the double difference estimation results.

## 5. Conclusions

Considering the importance of green finance to economic growth and ecological environment protection, we collected relevant data from the Yangtze River Delta (2010–2020). From the perspective of PSR index system, 12 indicators were selected to build an eco-environmental quality indicator system. The DID model was used to study the impact of green finance on the eco-environmental quality of the Yangtze River Delta. The conclusions are as follows.

First, the overall ecological environment quality of the YRD is on the rise. From the perspective of time sequence, the ecological environment quality of YRD showed an upward trend from 2010 to 2020. The comprehensive index of ecological environment quality increased from 0.4433 in 2010 to 0.6176 in 2020, with a growth rate of 39.3%. Environmental pollution has significantly improved, and the ecological environment has improved. In terms of YRD, the eco-environmental quality of the YRD showed an upward trend from 2010 to 2020. The growth rates of Shanghai, Jiangsu, Zhejiang and Anhui were 40.4%, 64.2%, 19.8% and 49.6%, respectively, among which Jiangsu had the largest increase, and the eco-environmental level improved significantly.

Second, green finance positively impacts the ecological and environmental quality of the Yangtze River Delta. The calculation results of the DID model show that green finance has a significant positive impact on improving regional ecological environment quality. It can optimize the allocation of financial resources, guide resources to enter more green environmental protection enterprises, reduce pollutant emissions, and enable green finance to improve the ecological environment quality.

However, this paper also has shortcomings. In the measurement of green finance, there is a lack of municipal data, so there is a lack of accuracy in the measurement of green finance. Secondly, this paper has few research objects, so it is difficult to analyze heterogeneity. Therefore, in the next stage, we will strengthen the collection and statistics of green finance data and build a more representative index system. We will take 34 provinces in China as the research object and use the threshold regression model for heterogeneity analysis. According to the characteristics of each region, the correlation between green finance and regional ecological environment will be further analyzed.

## 6. Countermeasures

The YRD plays an important leading role in the modernization of our country. The development of green finance can not only find new economic growth points for the YRD, but also be the key for China to practice the concept of green development. Based on the above results and discussion, the following suggestions are put forward:

First, strengthen the disclosure of green financial information. China’s green finance system is not perfect, and the information disclosure of green finance is not perfect. When commercial banks and financial institutions issue reports, all provinces and regions should also strengthen statistics of green finance data. The government and CBRC should not only supervise the development of green finance and disclose relevant information but also require financial institutions to carry out stricter examination and approval of green finance projects. Second, focus on the concept of green development. The development of green finance in the YRD significantly promotes the improvement of ecological environment quality. It is necessary to continue to increase investment in industrial pollution control and improve the industrial pollutant treatment capacity of the YRD. All YRD should pay close attention to green construction and ecological civilization construction. At the same time, economic growth and social development should adhere to a green and low-carbon cycle and promote harmony between man and nature. Third, strengthen regional cooperation and exchanges. There are certain differences in ecological environment quality and green finance development throughout the YRD. Shanghai, Jiangsu, Zhejiang, and Anhui need to strengthen communication and cooperation, understand their advantages and disadvantages, and promote the sharing of resources and technologies for the overall coordinated development. Regions with economic, human resource, and technological advantages can lead in driving the development of disadvantaged regions. Disadvantaged areas also need to speed up the introduction of talent and technology to promote the overall development of the YRD region.

## Figures and Tables

**Figure 1 ijerph-19-10692-f001:**
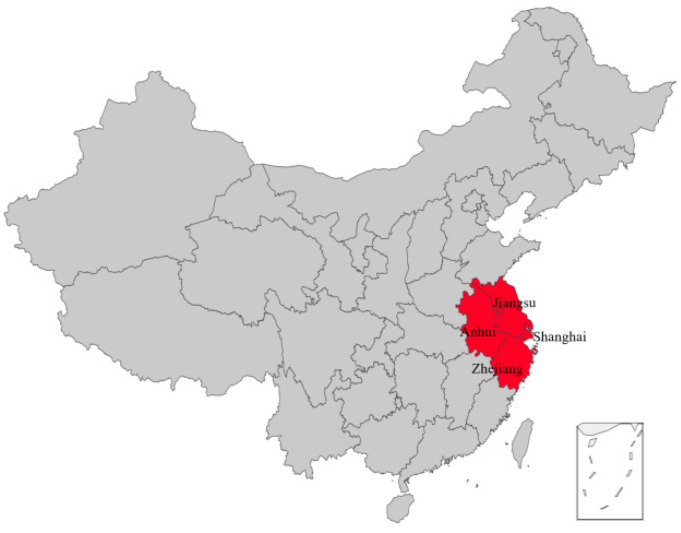
Yangtze River Delta map.

**Table 1 ijerph-19-10692-t001:** Index system of eco-environmental quality.

Level Indicators	Secondary Index (Unit)	Logo	Direction
Pressure	Industrial wastewater discharge (ten thousand tons)	X1	−
	Industrial smoke emission (ten thousand tons)	X2	−
	Sulfur dioxide emissions (ten thousand tons)	X3	−
	Production of general solid waste (ten thousand tons)	X4	−
State	Water resources per capita (m^3^/person)	X5	+
	Per capita urban park area (m^2^/person)	X6	+
	Road occupancy per capita (m^2^/person)	X7	+
	Forest coverage rate (%)	X8	+
	Green coverage rate of built-up area (%)	X9	+
Response	Urban domestic sewage treatment rate (%)	X10	+
	Comprehensive utilization rate of industrial solid waste (%)	X11	+
	Harmless treatment rate of household garbage (%)	X12	+

**Table 2 ijerph-19-10692-t002:** Variable description.

Variable Type	Indicator Name	Variable Interpretation	Abbreviation
Explained Variable	The level of ecological environment quality	The comprehensive index of ecological Environment quality	EQ
Explanatory Variable	Green finance	The investment amount in industrial pollution control	GF
Control Variables	The level of foreign opening	The proportion of total imports and exports in GDP	OPEN
	The level of economic development	GDP per capita	PGDP
	Industrial structure	The proportion of the added value of the secondary industry in GDP	STRU

**Table 3 ijerph-19-10692-t003:** Descriptive statistics of eco-environmental quality index evaluation system.

Indicator Name	Mean	Min	Max	Std. Dev.
Industrial wastewater discharge (ten thousand tons)	110,193.15	29,100.00	263,760.00	70,238.4926
Industrial smoke emission (ten thousand tons)	30.77	0.78	92.66	22.82
Sulfur dioxide emissions (ten thousand tons)	39.20	0.54	105.38	29.89
Production of general solid waste (ten thousand tons)	7890.00	1789.00	16,571.00	4823.30
Water resources per capita (m^3^/person)	1029.35	89.10	2644.80	792.50
Per capita urban park area (m^2^/person)	12.07	6.97	15.34	2.73
Road occupancy per capita (m^2^/person)	16.76	4.04	25.62	7.72
Forest coverage rate (%)	27.90	9.41	59.43	19.08
Green coverage rate of built-up area (%)	40.07	36.20	43.50	1.86
Urban domestic sewage treatment rate (%)	92.92	82.70	97.70	4.20
Comprehensive utilization rate of industrial solid waste (%)	91.55	79.83	99.01	4.72
Harmless treatment rate of household garbage (%)	96.10	61.00	100.00	8.57

**Table 4 ijerph-19-10692-t004:** Descriptive statistics of each variable.

Variables	Mean	Min	Max	Std. Dev.
EQ	0.521	0.262	0.839	0.169
GF	12.49	10.86	13.61	0.758
OPEN	0.577	0.112	1.413	0.369
PGDP	80,272.909	21,923	156,803	33,617.098
STRU	0.432	0.266	0.528	0.073

**Table 5 ijerph-19-10692-t005:** Level of eco-environmental quality in the YRD.

Year	Shanghai	Jiangsu	Zhejiang	Anhui	Mean Value
2010	0.2768	0.3399	0.6930	0.4637	0.4433
2011	0.2618	0.3613	0.6292	0.4728	0.4313
2012	0.3036	0.3550	0.7463	0.4846	0.4724
2013	0.3014	0.4098	0.7132	0.5189	0.4858
2014	0.3230	0.4376	0.7531	0.5516	0.5164
2015	0.3408	0.4686	0.7936	0.5766	0.5449
2016	0.3578	0.4844	0.7847	0.6071	0.5585
2017	0.3611	0.4835	0.7427	0.5969	0.5461
2018	0.3539	0.4949	0.7688	0.6142	0.5579
2019	0.3632	0.4744	0.8387	0.5618	0.5595
2020	0.3885	0.5582	0.8301	0.6937	0.6176
Mean value	0.3302	0.4425	0.7540	0.5584	0.5212

**Table 6 ijerph-19-10692-t006:** Sorting out panel regression results.

Variables	(1)		(2)	
	Coefficient	Std. Err.	Coefficient	Std. Err.
$treat_{i}\times time_{t}$	0.0237	−0.1103	0.0324 **	0.0034
GF	/	/	0.0231 **	0.0213
OPEN	/	/	0.2091 **	0.0027
PGDP	/	/	0.0112 ***	0.0000
STRU	/	/	−0.689 *	0.0913
Constant	−0.542	−0.172	0.322	0.0334

*** *p* < 0.01, ** *p* < 0.05, * *p* < 0.1.

**Table 7 ijerph-19-10692-t007:** Robustness test.

Variables	Coefficient	Std. Err.
$treat_{i}\times time_{t}$	0.0337	−0.2113
GF	0.0347 **	−0.0113
OPEN	0.102 *	−0.0722
PGDP	0.121 ***	0.000
STRU	−0.522	−0.153
Constant	0.126	−0.2112

*** *p* < 0.01, ** *p* < 0.05, * *p* < 0.1.

## Data Availability

Not applicable.

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
