# Peer review of "Influence of Green Finance on Ecological Environment Quality in Yangtze River Delta"

_ijerph, 2022, doi:10.3390/ijerph191710692_

Round 1
Reviewer 1 Report
In this paper, authors selected 12 indicators to construct the index system of eco-environmental quality and used panel data regression to analyze the effects of green finance on the index of eco-environmental quality in the Yangtze River Delta (YRD). Although it may be an important question, the quality of this research is much lower than the level of modern academic publication. The main comments are as follows:
Major comments:
1. The construction of index system of eco-environmental quality, the key dependent variable, is very arbitrary. Authors didn’t provide any discussion or explanation why their index of eco-environmental quality is a suitable index in this research. I don’t understand why those 12 indicators are important enough to represent the eco-environmental quality. They didn’t provide any explanation from existing literature, or economic theories.
2. The empirical analysis is too simple to meet the standards of an academic research: (1) The panel data regression cannot provide any information on causality, the method of DID would be a better choice to analyze the causal relationship between green finance and eco-environmental quality; (2) the total number of samples in the panel regression is only 44 (4 provinces, 11 years), which is too few to conduct a satisfied panel data analysis; (3) there is neither robustness test nor heterogeneity analysis after the main panel regression, which weakens the credibility of main results.
Minor comments:
1. It seems like some sentences were directly translated from Chinese, for example, In Page 3, “it can be found that scholars at home and abroad have carried out relatively mature studies in related fields. In terms of the importance of developing green finance, domestic scholars focus more on the development of financial institutions and sustainable economic development.” The whole paper looks like a direct translation from some student’s thesis or coursework.
2. The mathematical variables and formulas are not well edited in this paper. Authors should use formula editor to edit all the variables and formulas.
Author Response
Dear reviewer, thank you very much for your valuable comments and suggestions. According to your proposal, we have made a revision of the paper from several aspects, so as to get your approval.
Point 1: The construction of index system of eco-environmental quality, the key dependent variable, is very arbitrary. Authors didn’t provide any discussion or explanation why their index of eco-environmental quality is a suitable index in this research. I don’t understand why those 12 indicators are important enough to represent the eco-environmental quality. They didn’t provide any explanation from existing literature, or economic theories.
Response 1: Following your suggestions, we have explained the eco-environmental quality index system. Please see Line 209-227.
Point 2: The empirical analysis is too simple to meet the standards of an academic research: (1) The panel data regression cannot provide any information on causality, the method of DID would be a better choice to analyze the causal relationship between green finance and eco-environmental quality; (2) the total number of samples in the panel regression is only 44 (4 provinces, 11 years), which is too few to conduct a satisfied panel data analysis; (3) there is neither robustness test nor heterogeneity analysis after the main panel regression, which weakens the credibility of main results.
Response 2: According to your suggestions, we apply DID Model to analyze the causal relationship between green finance and ecological environment quality. Please see Line 266-299, 350-382. Secondly, because there are few samples in this study, and they are all in the Yangtze River Delta region, the differences between samples are small, and the effect of heterogeneous analysis is poor. In the next stage, the research team will take 34 provinces in China as the research objects and use the threshold regression model for heterogeneity analysis. Furthermore, study the promotion effect of Green Finance on the improvement of regional ecological environment quality. Finally, the robustness test is added. Please see Line 385-395.
Point 3: It seems like some sentences were directly translated from Chinese, for example, In Page 3, “it can be found that scholars at home and abroad have carried out relatively mature studies in related fields. In terms of the importance of developing green finance, domestic scholars focus more on the development of financial institutions and sustainable economic development.” The whole paper looks like a direct translation from some student’s thesis or coursework.
Response 3: According to your suggestion, we have checked and improved the grammar of the full text.

Reviewer 2 Report
In this manuscript, the authors study the Yangtze River Delta green financial impact on the quality of the ecological environment and have certain innovations in terms of the research object. However, I will comment on some aspects to improve the quality of the manuscript. In addition, the changes made by the authors should be highlighted.
-Some acronyms are incorrect. Acronyms must be written as presented in line 55. This error must be fixed throughout the document.
-Authors must avoid the use of phrasal verbs in a scientific article.
-The authors do not present a brief introduction of the Sections that the manuscript will have at the end of the Introduction or Related Works Section.
-There are objects such as Tables, Figures, and Equations that have not been correctly cited in the manuscript.
-The word "formula" must not refer to the Equation.
-Words such as Section, Figure, Algorithm, Table, and Equation must be written with their first capital letter.
-After line 243, there is a wasted space that must not have a scientific article.
-Where have the authors' data for the study been proposed?
-The title of Section 5 must only be Conclusions.
-The Conclusions Section must reduce and specify the conclusions obtained in the study. If the authors wish to discuss, a Discussion Section can be added.
-The authors have not added the Future Works in the Conclusions Section.
Author Response
Dear reviewer, thank you very much for your valuable comments and suggestions. According to your proposal, we have made a revision of the paper from several aspects, so as to get your approval.
Point 1: Some acronyms are incorrect. Acronyms must be written as presented in line 55. This error must be fixed throughout the document.
Response 1: We have checked and fixed the acronyms in the full manuscript according to your suggestion.
Point 2: Authors must avoid the use of phrasal verbs in a scientific article.
Response 2: We have checked and revised the full manuscript according to your suggestion.
Point 3: The authors do not present a brief introduction of the Sections that the
Response 3: According to your suggestion, we have added a brief introduction to the manuscript in the introduction. Please see Line 84-88.
Point 4: There are objects such as Tables, Figures, and Equations that have not been correctly cited in the manuscript.
Response 4: According to your suggestion, we have checked and fixed the Tables, Figures, and Equations in the full manuscript.
Point 5: The word "formula" must not refer to the Equation.
Response 5: According to your suggestion, we have checked and revised it.
Point 6: Words such as Section, Figure, Algorithm, Table, and Equation must be written with their first capital letter.
Response 6: According to your suggestion, we have checked and revised it.
Point 7: After line 243, there is a wasted space that must not have a scientific article.
Response 7: According to your suggestion, we have checked and revised it.
Point 8: Where have the authors' data for the study been proposed?
Response 8: Thank you for your suggestions. We have improved the description of the research data. Please see Line 300-312.
Point 9: The title of Section 5 must only be Conclusions.
Response 9: According to your suggestion, we have revised section 5. Please see Line 396-425.
Point 10: The Conclusions Section must reduce and specify the conclusions obtained in the study. If the authors wish to discuss, a Discussion Section can be added.
Response 10: We have revised the Conclusions Section according to your suggestion. Please see Line 396-425.
Point 11: The authors have not added the Future Works in the Conclusions Section.
Response 11: According to your suggestion, we have added Future Works in the Conclusions Section. Please see Line 418-425.

Reviewer 3 Report
This is a good research paper which gives enough context and background, provides good enough details on the methods, and then uses them to arrive at justified conclusions. Some of the introduction can be tightened to give some more scope on the regional nuances (something like a map would help), and the results can be illustrated a bit more with implications on external validity. The writing in general is good but the English can be improved a bit, especially in the lengthy text sections (introduction, background and conclusions).
Author Response
Dear reviewer, thank you very much for your valuable comments and suggestions. According to your proposal, we have made a substantial revision of the paper from several aspects, so as to get your approval.
Point 1: This is a good research paper which gives enough context and background, provides good enough details on the methods, and then uses them to arrive at justified conclusions. Some of the introduction can be tightened to give some more scope on the regional nuances (something like a map would help), and the results can be illustrated a bit more with implications on external validity. The writing in general is good but the English can be improved a bit, especially in the lengthy text sections (introduction, background and conclusions).
Response 1: First of all, according to your suggestion, we have added the Yangtze River Delta regional map so that some introductions in this article can be more rigorous and provide more scope in terms of regional nuances. Secondly, the research results are explained in more detail. Please see Line 336-380. Finally, the language is improved, especially in the length of text sections (introduction, background and conclusions).

Round 2
Reviewer 1 Report
The revised manuscript has been improved a lot and I don't have any major comment.
Author Response
Thank you very much for your approval.
Reviewer 2 Report
The authors have not performed all the changes suggested by the authors. In addition, the presentation of the manuscript for the authors is not adequate and indicates the low quality of the article, primarily due to the gray box on the right side of the article, the strikethroughs in the text and the different colors in the manuscript. Authors are asked to improve the presentation for reviewers by writing the final form and highlighting what has been changed in the article.
Author Response
Dear reviewer, thank you very much for your valuable comments and suggestions. According to your suggestion, we have improved the manuscript and highlighting the modified content. The modification details are as follows:
Point 1: Some acronyms are incorrect. Acronyms must be written as presented in line 55. This error must be fixed throughout the document.
Response 1: We have checked and fixed the acronyms in the full manuscript according to your suggestion. Please see Lines 48, 85, 115, 124, 133, 135, 136, 247, 277.
Point 2: Authors must avoid the use of phrasal verbs in a scientific article.
Response 2: We have invited a native English speaker with Ph.D. to check and revise the full manuscript according to your suggestion.
Point 3: The authors do not present a brief introduction of the Sections that the manuscript will have at the end of the Introduction or Related Works Section.
Response 3: According to your suggestion, we have added a brief introduction to the manuscript in the introduction. Please see Lines 84-89.
This paper attempts to study the impact of green finance on the ecological environment quality in the YRD. Based on the index system of Pressure State Response (PSR), considering the scientific nature and desirability of the indicators, we select 12 indicators to construct the index system of eco-environmental quality and use the Entropy Method to calculate the level of eco-environmental quality. Then, three control variables are selected, and the Difference-in-Difference Model is used for empirical analysis.
Point 4: There are objects such as Tables, Figures, and Equations that have not been correctly cited in the manuscript.
Response 4: According to your suggestion, we have checked and fixed the quotations of Tables, Figures, and Equations throughout the manuscript. Please see Lines 198-199, 227-228, 261, 267, 282-284.
Point 5: The word "formula" must not refer to the Equation.
Response 5: According to your suggestion, we have checked and revised it. Please see Lines 227-228, 261,
Point 6: Words such as Section, Figure, Algorithm, Table, and Equation must be written with their first capital letter.
Response 6: According to your suggestion, we have checked and revised it. Please see Lines 139, 206, 224, 289, 311.
Point 7: After line 243, there is a wasted space that must not have a scientific article.
Response 7: According to your suggestion, we have checked and revised it.
Point 8: Where have the authors' data for the study been proposed?
Response 8: Thank you for your suggestions. We have improved the description of the research data. Please see Lines 279-284.
This paper selects data from Shanghai, Jiangsu, Zhejiang, and Anhui within the YRD from 2010–2020, which can be acquired from the China Statistical Yearbook, China Statistical Yearbook on Environment, China City Statistical Yearbook, and the Ecological Environmental Status Bulletin. Table 3 reports descriptive statistical data of each indicator in the ECO-environmental quality indicator system. Table 4 reports descriptive statistics for variables in the DID Model.
Point 9: The title of Section 5 must only be Conclusions.
Response 9: According to your suggestion, we have revised section 5. Please see Lines 356-384.
- Conclusions
Considering the importance of green finance to economic growth and ecological environment protection, we collected relevant data from the Yangtze River Delta (2010-2020). From the perspective of PSR index system, 12 indicators are selected to build an eco-environmental quality indicator system. The DID Model is used to study the impact of Green Finance on the eco-environmental quality of the Yangtze River Delta. The conclusions are as follows.
First, the overall ecological environment quality of the YRD is on the rise. From the perspective of time sequence, the ecological environment quality of YRD showed an upward trend from 2010 to 2020. The comprehensive index of ecological environment quality increased from 0.4433 in 2010 to 0.6176 in 2020, with a growth rate of 39.3%. Environmental pollution has significantly improved, and the ecological environment has improved. In terms of YRD, the eco-environmental quality of the YRD showed an upward trend from 2010 to 2020. The growth rates of Shanghai, Jiangsu, Zhejiang, and Anhui were 40.4%, 64.2%, 19.8% and 49.6%, respectively, among which Jiangsu had the largest increase, and the eco-environmental level improved significantly.
Second, green finance positively impacts the ecological and environmental quality of the Yangtze River Delta. The calculation results of the DID Model show that green finance has a significant positive impact on improving regional ecological environment quality. It can optimize the allocation of financial resources, guide resources to enter more green environmental protection enterprises, reduce pollutant emissions, and enable green finance to improveme the ecological environment quality.
However, this paper also has shortcomings. In the measurement of green finance, there is a lack of municipal data, so there is a lack of accuracy in the measurement of green finance. Secondly, this paper has few research objects, so it is difficult to analyze heterogeneity. Therefore, in the next stage, we will strengthen the collection and statistics of green finance data and build a more representative index system. We will take 34 provinces in China as the research object and use the threshold regression model for heterogeneity analysis. Further analyse the correlation between green finance and regional ecological environment according to the characteristics of each region.
Point 10: The Conclusions Section must reduce and specify the conclusions obtained in the study. If the authors wish to discuss, a Discussion Section can be added.
Response 10: According to your suggestion, we have revised the Conclusions Section. Please see Lines 355-384. We added the discussion section to the section 4. Please see Lines 300-309, 321-342.
Point 11: The authors have not added the Future Works in the Conclusions Section.
Response 11: According to your suggestion, we have added Future Works in the Conclusions Section. Please see Lines 381-385.
Therefore, in the next stage, we will strengthen the collection and statistics of green finance data and build a more representative index system. We will take 34 provinces in China as the research object and use the Threshold Regression Model for heterogeneity analysis. According to the characteristics of each region, the correlation between green finance and regional ecological environment will be further analysed.
